# Predictors for psychosocial consequences of screening for liver diseases: A data-driven approach

Urs Alexander Fichtner[1]*, Andy Maun[2], Erik Farin-Glattacker[1]

1 Institute of Medical Biometry and Statistics, Section of Health Care Research and Rehabilitation Research, Faculty of Medicine and Medical Center – University of Freiburg, Freiburg im Breisgau, Germany, 2 Department for General Practice, Faculty of Medicine and Medical Center – University of Freiburg, Freiburg im Breisgau, Germany

* Urs.Fichtner@uniklinik-freiburg.de

## Abstract

### Background

Medical screening is employed to detect early signs of diseases in asymptomatic populations, potentially improving patient outcomes through early intervention. However, the psychosocial impact of screenings remains a field of discussion. Inconsistent findings from studies, mainly originally from cancer research, are not easily transferable to the context of liver screening. This study aimed to identify predictors of psychosocial consequences in asymptomatic adults screened for early-stage liver cirrhosis, thereby contributing to the current knowledge on screening impact.

### Methods

We analyzed data from 487 participants who underwent a systematic liver disease screening in Germany from January 2018 to February 2021. The screening involved blood tests, advanced diagnostics, and potentially, liver biopsies. We used bootstrapped LASSO regression with 10-fold validation to evaluate the influence of various predictors on psychosocial outcomes measured by the Psychological Consequences of Screening Questionnaire (PCQ).

### Results

The results show that severity of comorbidities (beta = 0.44–2.72), subjective social status (beta = −0.30–−0.86), and social support (beta = −0.33–−0.98) were consistent predictors across all psychosocial outcome measures by not covering zero in the confidence intervals. Older age (beta = −0.03–−0.08), the existence of a steady partnership (beta = −1.08–−0.48) and higher health literacy regarding the application of medical information (beta = 0.33–0.48) were associated with less psychosocial dysfunction, indicating their protective roles to prevent psychosocial burden of screening.

**Data availability statement:** Data used for this analysis was published on ZENODO: Fichtner, U. (2024). SEAL Substudy dataset on psychosocial consequences of liver screening (Version V2) [Data set]. Zenodo. doi:10.5281/ZENODO.113946021

**Funding:** The data used for this Study was collected within a project (SEAL Programme) funded by the Innovation Fund of Federal Joint Committee – Gemeinsamer Bundesausschuss (G-BA) Germany registered under the funding ID: 01NVF16026 (Awarded to EFG). The funders had no role in study design, data collection and analysis, decision to publish, or preparation of the manuscript. We acknowledge support by the Open Access Publication Fund of the University of Freiburg.

**Competing interests:** The authors have declared that no competing interests exist.

## Conclusion

The study underscores the importance of considering individual patient characteristics in predicting psychosocial consequences of medical screening. Medical practitioners should consider personalized communication strategies taking into account the individual context of patients. The protective role of social support and stable personal relationships suggests that integrating psychosocial support services within screening programs could mitigate negative outcomes. Furthermore, increasing patient health literacy might help to demystify the screening process and can reduce psychosocial burden even if patients come from a segment of lower subjective social status.

## Introduction

Medical screening is performed on members of an asymptomatic population to assess their likelihood of having a particular disease [1]. Screening programs are increasingly used in any medical discipline to detect the onset of diseases and prevent severe progression [2]. By this, medical screening has the potential to reduce disease-specific and overall mortality and to improve quality of life [3]. However, screening repeatedly faces discussions since its overall benefit cannot be carved out clearly. An epidemiologic review from 2011 by Harris points out that medical screening is generally of complex evidence [4]. This leads to confusion among clinicians and policy makers and limits a broad acceptance of screening procedures [5]. On the one hand, several benefits of early disease detection through screening were observed. In the context of infectious diseases, for example, screening is found to reduce risky sexual behaviors and transmission of HIV [5]. Improved health behavior in terms of prevention is evident also in the context of cancer screening [6–8]. From public health perspective, studies in the field of cancer screening often conclude that screening is cost-effective or even cost-saving compared to no screening, expressed in quality-adjusted life year gains [7,9]. Nevertheless, many of these studies examining cost-effectiveness concentrate exclusively on economic costs measured by the expenditures for the organization of screening programs or avoided treatment costs by early detection of diseases [10,11].

From a holistic perspective, however, also non-economic costs have to be taken into account. The major harm that is associated with screenings in general are psychosocial adverse effects. Psychological distress is repeatedly reported in the context of screenings, for, e.g., diabetes [12], breast [13], anal [14], colorectal [15] and skin cancer screening [8]. The participation in screening programs is found to be associated with fears [16–18], anxiety [13,19,20], sadness [21], sleeping disorders [22], and physical dysfunction [21,23].

Furthermore, medical screening bears the risk of overdiagnosis, which is found to be the most important harm [4]. Overdiagnosis refers to the detection of a disease that, if left undetected, might not cause symptoms or death during a patient's lifetime. The harms of overdiagnosis for screened patients' psychosocial well-being might be

very similar to those that arise in case of false-positive screening results. Latter occur, when a screening test result is positive, even though the detected disease is not apparent. In both cases, the reception of the diagnosis can be considered unnecessary or even harmful. A longitudinal study on consequences of false-positive screening mammograms concludes that false-positive mammograms were associated with increased short-term anxiety, though long-term anxiety was not different across the two observed groups [24]. The appearance of short-term distress, but no remaining long-term anxiety or depression was confirmed in another study on false-positive mammograms [25]. In contrast, Brodersen & Siersma found that false-positive findings on screening mammography had a long-term psychosocial impact even after a period of three years [22]. Another study from Denmark, that observed women with false-positive mammogram results over a follow-up period of 14 years, supports these findings [26]. In a study on a colorectal cancer screening program in Denmark, both short- and long-term consequences lasting for at least one year were found for patients even after receiving the diagnosis of no abnormalities [27]. In the context of HIV screening, false positive results, which remain unrectified for a longer period, were found to lead to psychiatric morbidities such as depression, panic attacks and suicidal thoughts [28]. Ong et al contribute to the discussion by concluding that the mode and time point, when test results are communicated, influence the level of adverse psychological consequences [13]. Supplementary, the high variation in the results of studies investigating psychosocial consequences of screening can partly be explained by whether a generic or a specific outcome was investigated [29].

Based on the very heterogeneous results and contexts reported in the literature, it is difficult to generalize findings about negative consequences of medical screening, regardless whether results were (false-)positive or not. The main body of the studies is focusing on variants of cancer screening and it is not clear whether these findings are transferable to other contexts of medical screening, e.g., for liver cirrhosis. The literature differs between measured outcome (general vs. disease-specific psychosocial outcomes), different foci (false-positive results vs. screening results in general), observational period (short vs. long-term outcomes) and of course in the investigated study population (e.g., only women in mammography screening and only men in prostate cancer screening). Hence, since psychosocial consequences of screening seem to vary in terms of their extent, magnitude and duration, it is barely possible to identify general patient-inherent characteristics that can predict them.

In the context of liver screening, the consequences of false-positive screening results and overdiagnosis can be considered similar to the psychosocial consequences of having a confirmed diagnosis of liver cirrhosis or fibrosis. A scoping review identified a set of 152 patient-reported outcome measures (PROMs) that were reported by patients with liver cirrhosis [30]. Besides physical functioning and symptoms, the domains also cover mental health, cognition, social life and satisfaction with care and thus relate to psychosocial outcomes of having liver cirrhosis. The affected patients suffer from impaired quality of life and from psychosocial burden on close relatives resp. informal caregivers and experience financial burden, especially when socioeconomic status is low [31–33].

Within the SEAL program (Structured Early detection of Asymptomatic Liver cirrhosis) a systematic screening procedure to detect patients with liver cirrhosis at early stage in an asymptomatic general population was evaluated [34,35]. Besides a cost-effectiveness analysis [11] and an overall feasibility study [34], one aim of the SEAL program was to explore the psychosocial outcomes of the screening. Using a qualitative approach, we conducted semi-structured telephone interviews with 11 positively screened patients [36]. The results were heterogeneous. While some respondents described negative emotional consequences related to the screening itself, others did not report any negative feelings. Furthermore, we wanted to investigate patient-inherent characteristics that predict and explain variance in psychosocial outcomes of (positively) screened patients. For this reason, the present survey study was designed with the following aims:

1. Determining predictors that explain variance in post-screening psychosocial consequences in the context of a liver screening for cirrhosis and fibrosis.

2. Identifying variables that negatively impact psychosocial outcomes.

3. Finding protective variables to reduce negative psychosocial outcomes.

## Materials and methods

### Study design

Within the SEAL program, patients who visited collaborating clinics or doctor's offices in Rhineland-Palatinate or Saarland, Germany, for a regular check-up were screened for liver cirrhosis and fibrosis in the time frame from January 2018 to February 2021. The screening protocol is designed as a series of steps, with each stage proceeding based on the results of the previous test. Firstly, a blood sample test is taken and a risk score (Aspartate aminotransferase to Platelet Ratio (APRI) with a cut-off value of 0.5) was determined (Step 1) [37]. In case of an increased APRI, more sophisticated laboratory diagnostics and ultrasound imaging followed (Step 2). The final stage involves conducting a liver biopsy and advanced diagnostics within a specialized medical facility (Step 3). Eligibility for participation in the study was restricted to individuals aged 35 and above, with no prior diagnosis of liver cirrhosis [35].

### Participants and recruitment

In August 2019, we contacted all participants who had been included in the study up to that point. Data security constraints prevented us from determining the specific phase in which each patient was within the screening process. Consequently, we were unable to ascertain whether individuals with positive initial screenings had progressed to the subsequent steps or were still awaiting further appointments. This limitation hindered our ability to account for the impact of false-positive findings and their potential consequences. Given that the screening initiative commenced in January 2018, it is reasonable to assume that most participants had been informed about their screening outcomes by this time. Nevertheless, insights from an earlier qualitative study indicated that patients typically do not receive any communication regarding negative screening outcomes, meaning those test steps which did not identify any disease indicators [36].

The recruitment time point was determined based on a simple sample size calculation. We aimed to survey at least n = 100 positively screened patients to compare this group with negatively screened patients. Assuming a positive screening rate of 4–5% and a return rate of 40%, we chose a date by which approximately 6,000 individuals were included in the SEAL program.

### Data collection

A total of 5,935 screening participants were sent a postal mailing that contained a paper-based self-administered questionnaire along with patient information, an informed consent form and a preaddressed return envelope. We produced two versions of the questionnaire with a slightly enhanced module for the positively screened patients to evaluate the subsequent steps they have undergone in the screening procedure. Where possible, we used standardized instruments and complemented the questionnaire by self-developed items based on insights gained from qualitative patient interviews conducted in advance [36]. The instrument encompasses a context-adapted German version of the Psychological Consequences of Screening Questionnaire (PCQ) [2,6], the short form of the State-Trait Anxiety Inventory (STAI) [38], a multi-morbidity score (KOMO) [39], a health literacy instrument (HELP) [40], the Oslo Social Support Scale (OSSS-3) [41], the short form for health-related quality of life (SF-12) [42], items to measure communication competences (KoKo) [43], items on symptom changes [44], the MacArthur Scale on Subjective Social Status [45] and items on satisfaction with healthcare (ZAP) [46]. The self-developed part includes items on the communication of the screening test result, satisfaction with information on screening and care, future screening attitudes, sex, age, cohabitation, marital status, education, and occupational status. Some of them were derived from the background variables questionnaire of the International Social Survey Programme (ISSP) [47]. The PCQ aims to cover three constructs of psychosocial consequences of screening: emotional dysfunction, physical dysfunction and social dysfunction. The processed data used for this analysis was published on ZENODO [48].

## Statistical procedure

To identify relevant predictors for psychosocial consequences, we applied LASSO (least absolute shrinkage and selection operator) regressions using the R package glmnet [49]. By adding a penalty equal to the absolute value of the magnitude of coefficients, this method is preferable in comparison to other regressive methods to perform variable selection. By shrinking coefficients to zero, it helps to produce simpler models. Furthermore, it can mitigate multicollinearity better than traditional regression methods [50]. As dependent variables, we included the PCQ overall score, as well as its three subscales for physical, social and emotional dysfunction. As potential predictors, we included calculated scores and indices, where it was suggested in the literature, or the raw variables (see Table 1).

To enhance variable selection consistency, we produced 100 bootstrapped data samples from our survey data for each of the four dependent variables and run the LASSO models on them. This procedure is recommended to reduce variability in variable selection and to increase robustness of our results [51]. For selection of the finalized (minimal) Lambda, we used 10-fold cross-validation [52]. To assess model performance, we calculated the average performance across all bootstrap samples using the root mean square error (RMSE) and the mean absolute error (MAE). If the RMSE was lower than the standard deviation of the dependent variable, we considered the LASSO models as well fitted on average [53]. As cutoff for important variable selection, we decided a 80% threshold meaning that the predictor of interest had to be considered relevant in at least 80% of the bootstrapped models. This strict cut-off was chosen in order to identify predictors that are as reproducible as possible. As final step, we extracted the regression coefficients of the relevant predictors for each of the four models. For assessing the significance of the found predictors, we calculated 95% confidence intervals (CI) for the averaged coefficients across all bootstrapped samples [52].

## Ethics statement

This study was reviewed and approved by Ethics Committees of Rhineland-Palatinate and Saarland, Germany. The participants were informed about the legal base for data collection and procession and we only operated data when written informed consent for study participation was given.

## Results

### Sample description

With a return rate of 9% in those who were negatively screened and a return rate of 12% in those who were positively screened, our database for the analysis includes n = 502 questionnaires from participants who were negatively screened and n = 21 questionnaires from participants who were positively screened (gross sample n = 523). After exclusion of those cases that were not legitimate for evaluation (n = 36) due to missing informed consent or dubious response, we ended up with a net sample of n = 487 (n = 19 positively screened and n = 468 negatively screened). Table 2 shows that gender is almost equally represented in our sample. The mean age was 62 years closely aligning with the SEAL cohort's average age of 61 years. 48% of the respondents were retired, while 42% were employed. The majority of the sample (64%) had primary or lower secondary school degree and 58% had vocational training. On average, respondents lived in a 2-person household and three quarters of them had a steady partner. The average subjective social status was 7, which lies near to representative German studies [54]. A third (31%) of our sample reports poor social support. Regarding the screening-specific variables, we achieved a good representation of the SEAL cohort (4,12%, [34]) with 4% positively screened patents. Less than 10% of the sample had little or no trust in their general practitioners, rated the quality of treatment low or were (rather) dissatisfied with their medical carers. More than 60% of the sample either agreed or agreed strongly to the items related to information satisfaction and to feeling of being in good hands. 88% of the respondents had already received their test results at the time point of filling out the questionnaire. Less than half of the sample planned to participate in a liver screening within the next two years. Regarding communication competences, the mean of our sample

**Table 1. Variables used for LASSO regression.**

| Variable | Source | Range |
|---|---|---|
| *Outcomes* | | |
| Psychosocial Consequences (Overall score) | PCQ negative | [0–36] |
| Emotional dysfunction | PCQ negative | [0–15] |
| Physical dysfunction | PCQ negative | [0–12] |
| Social dysfunction | PCQ negative | [0–9] |
| *Potential predictors* | | |
| critical and participatory communication index | KoKo | [0–100] |
| active and disease-specific communication index | KoKo | [0–100] |
| Difficulties applying medical information (index) | HELP | [1–5] |
| Difficulties interacting with medical staff (index) | HELP | [1–5] |
| Number of comorbidities Score | KOMO | [0-12] |
| Standardized Severity Score | KOMO | [0-10] |
| Trust in general practitioner | ZAP item | [1–4] |
| Quality of treatment | ZAP item | [1–4] |
| Satisfaction with general practitioner | ZAP item | [1–4] |
| Social Support index | OSSS-3 | [1–3] |
| Subjective social status | MacArthur Scale | [1–10] |
| screening outcome: positive or negative | administrative | [0–1] |
| Satisfaction with information on screening procedure | self-developed | [1–5] |
| Satisfaction with information on risk factors | self-developed | [1–5] |
| Feeling of being in good hands: general practitioner | self-developed | [1–5] |
| Household size (persons) | Single item, ISSP | [1–7] |
| Education: school degree | Single item, ISSP | [1–5] |
| Education: professional training | Single item, ISSP | [1–4] |
| Occupational status, recoded | Single item, ISSP | [0–1] |
| Existence of a life partner | Single item, ISSP | [0–1] |
| Received information of test result | self-developed | [0–1] |
| future participation in liver screening | self-developed | [0–1] |
| Sex | Single item, ISSP | [0–1] |
| Age (years) | Single item, ISSP | [31–98] |

*Note. We did not include health-related quality of life (SF-12), anxiety (STAI) and items on symptoms change since these constructs are very similar to the PCQ. Scale ranges: PCQ scales: higher values indicate higher dysfunction. KoKo: higher values indicate higher competence. HELP: higher values indicate higher difficulties. KOMO: higher values indicate more comorbidities and higher severity. ZAP-items: 1 "very much/very high/very satisfied", 2 "rather much/rather high/rather satisfied" 3"rather little/rather dissatisfied", 4 "no trust at all/very little/very dissatisfied". STAI: higher values indicate higher anxiety. OSSS-3: 1 "poor", 2 "moderate", 3 "strong" support. MacArthur: higher values indicate higher subjective social status. Satisfaction with information/good hands: 1 "disagree totally", 2 "disagree", 3 "neutral", 4 "agree", 5 "agree totally". Education: school degree: 1 "none (yet)", 2 "CSE/Primary", 3 "GCSE", 4 "Technical college", 5 "High school diploma". Professional training: 1 "none", 2 "vocational training", 3 "specialized academy, professional school", 4 "university, technical college". Occupational status: 0 "retired/househusband/-wife/unemployed/permanently unable to work", 1 "employed/in education". Binary coded variables: 0 "no/not present" 1 "yes/present".*

lay 10 points above the scale mean indicating moderate communication competences. Difficulties in applying medical information (1.78) and in interacting with medical staff (1.82) were small on average. The mean number of comorbidities was relatively high with four diseases reported; however, the mean degree of severity was in the lower fifth of the scale. Mean anxiety was 39 which represents middle anxiety on the standardized scale.

**Table 2. Sample characteristics.**

| | Sample statistics | | |
|---|---|---|---|
| | % | Mean | Standard deviation |
| ***Outcome variables*** | | | |
| Psychosocial Consequences (PCQ) | | | |
| Total score | | 6.44 | 8.61 |
| Emotional dysfunction | | 3.26 | 4.43 |
| Social dysfunction | | 1.24 | 2.05 |
| Physical dysfunction | | 2.27 | 3.08 |
| ***Health-related variables*** | | | |
| Communication competences (KoKo) | | | |
| Critical and participatory | | 60.57 | 21.61 |
| Active and disease-specific | | 58.33 | 24.30 |
| Health Literacy (HELP) | | | |
| Difficulties applying medical information | | 1.78 | 0.75 |
| Difficulties interacting with medical staff | | 1.82 | 0.87 |
| Comorbidities (KOMO) | | | |
| Number of comorbidities | | 4.03 | 2.52 |
| Standardized severity | | 1.73 | 1.26 |
| Anxiety (STAI) | | 39.13 | 13.14 |
| ***Screening-related variables*** | | | |
| Screening test result | | | |
| Negative | 96.1 | | |
| Positive | 3.9 | | |
| Treatment evaluation (ZAP) | | | |
| Trust in GP (little, no trust) | 4.9 | | |
| Quality of treatment (rather little, very little) | 6.5 | | |
| Satisfaction with GP ((rather) dissatisfied) | 4.4 | | |
| Information satisfaction | | | |
| Screening procedure ((strongly) disagree) | 11.8 | | |
| Risk factors ((strongly) disagree) | 15.7 | | |
| Feeling of being in good hands ((strongly) disagree) | 6.1 | | |
| Reception of test result (yes) | 88.2 | | |
| Future participation in liver screening (yes) | 46.4 | | |
| ***Sociodemographics*** | | | |
| Social Support (OSSS-3) | | | |
| Poor | 31.4 | | |
| Moderate | 48.3 | | |
| Strong | 20.2 | | |
| Subjective Social Status (MacArthur) | | 7.00 | 1.70 |
| Household size (persons) | | 2.28 | 1.07 |
| Life partner (yes) | 77.5 | | |
| Highest school degree | | | |
| Primary/ lower secondary | 63.7 | | |
| Professional education | | | |
| Vocational training | 58.5 | | |
| Occupation | | | |
| Employed | 41.9 | | |

*(Continued)*

**Table 2.** (Continued)

| | Sample statistics | | |
|---|---|---|---|
| Retired | 48.1 | | |
| Age | | 62.37 | 12.26 |
| Sex | | | |
| Male | 49.2 | | |
| Female | 50.8 | | |

Regarding the outcome values of interest, we found low overall dysfunction on the total score with 6.44 on average. For social, emotional and physical dysfunction, the mean values were also beyond the lower fourth bound of the scale, indicating little dysfunction in consequence of the screening. The PCQ and its subscales were not normally distributed which supports the decision to apply bootstrapping to calculate a linear model. In a bivariate Mann-Whitney-U-test, we compared the total PCQ across the two screening groups and found a significant difference on a 90% significance level (Test statistic 4580, p = 0.066) with higher mean dysfunction in the positively screened group (9.71) than in the negatively screened group (6.31).

## Variable selection

We run the LASSO models on all four dependent variables of interest. Fig 1 displays the included predictors for each set of models and ranks them by frequency of identification as relevant for the 100 bootstrapped data samples per outcome. All four models have in common that three variables exceeded the 80% threshold and thus, were identified as relevant predictors for every predicted outcome: severity of comorbidities (KOMO), subjective social status (MacArthur) and social support (OSSS-3). Age was identified as relevant for the total PCQ, the emotional and the social dysfunction subscale. The existence of a steady life partner, which might also be considered as proxy for social support, was identified as important for all subscales, but not the total scale. One of the two health literacy subscales (HELP) was also identified as potential predictors in the emotional and the social subscale. Here, the subscale for difficulties in application of health information reached the 80% threshold while the subscale for interaction with healthcare providers did not. Overall, the selected variables are quite homogeneous across the models and no further predictors were identified.

## Final models

The results from the regularized LASSO regressions with selected predictors are displayed in Table 3. The unstandardized regression coefficients and standard errors represent averaged values across all 100 bootstraps for each model type. A higher degree of comorbidity severity was correlated with higher dysfunction on all four outcomes and the CI does not include 0 in any model, indicating a significant effect. This effect is plausible, since some of the items of the PCQ reflect the severity of comorbidities, e.g., the PCQ item phrasing, e.g., "I felt under strain".

Concerning the protective predictors for psychosocial consequences of screening, we found that higher subjective social status was associated with lower scale values on the PCQ. Here, the CIs also lie away from 0 and the standard errors are quite small. This finding is plausible, since subjective social status was repeatedly found to be associated with differences in health outcomes [55]. This predictor was considered relevant, regardless of the included dependent variable.

Social support played also an important role as protective factor for lower psychosocial consequences of screening. This social support effect is also apparent by the existence of a steady partnership. In all three subscale models, having a steady partnership also decreased psychosocial consequences. This result is not surprising, since social support is found

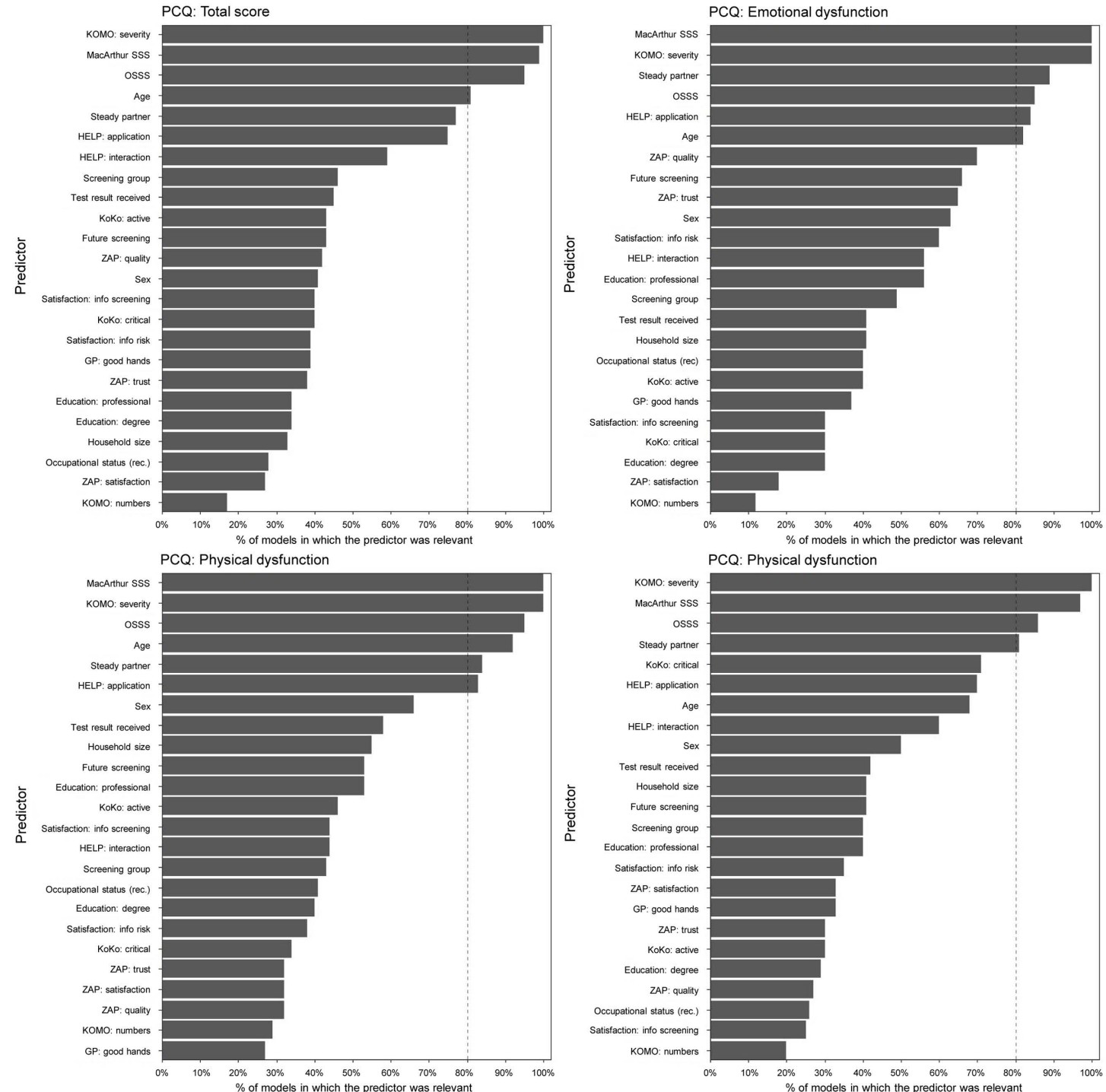

**Fig 1. Overview of selected variables (cut-off-value of 80%) for PCQ total, emotional, social and physical dysfunction.**

**Table 3. Final LASSO regression models on the dependent variables (PCQ total, emotional, social and physical subscale) after variable selection (80% cut-off).**

| Predictors | Dependent variable | | | | | | | | | | |
|---|---|---|---|---|---|---|---|---|---|---|---|
| | PCQ total | | | Emotional dysfunction | | | Social dysfunction | | | Physical dysfunction | | |
| | regression coefficient | standard error | 95% CI | regression coefficient | standard error | 95% CI | regression coefficient | standard error | 95% CI | regression coefficient | standard error | 95% CI |
| Severity of comorbidities | 2.72 | 0.066 | [2.59;2.85] | 1.47 | 0.032 | [1.41;1.53] | 0.44 | 0.017 | [0.41;0.48] | 0.94 | 0.022 | [0.90;0.98] |
| Subjective social status | −0.86 | 0.029 | [−0.91;−0.80] | −0.30 | 0.012 | [−0.32;−0.28] | −0.24 | 0.008 | [−0.26;−0.23] | −0.29 | 0.010 | [−0.31;−0.27] |
| Social support | −0.98 | 0.051 | [−1.07;−0.88] | −0.33 | 0.024 | [−0,37;−0,29] | −0.24 | 0.015 | [−0.27;−0.21] | −0.26 | 0.015 | [−0.29;−0.23] |
| Age | −0.08 | 0.005 | [−0.08;−0.07] | −0.03 | 0.002 | [−0.04;−0.03] | −0.03 | 0.002 | [−0.03;−0.02] | | | |
| Steady partnership | | | | 1.08 | 0.069 | [−1.21;−0.95] | −0.48 | 0.031 | [−0.54;−0.43] | −0.59 | 0.039 | [−0.66;−0.52] |
| Health literacy: application | | | | 0.48 | 0.040 | [0.40;0.55] | 0.33 | 0.021 | [0.30;0.37] | | | |
| **Model fit** | | | | | | | | | | | | |
| RMSE | 6.74 | | | 3.38 | | | 1.77 | | | 2.42 | | |
| Standard deviation of scale | 8.70 | | | 4.35 | | | 2.36 | | | 3.07 | | |
| MAE | 5.24 | | | 2.64 | | | 1.33 | | | 1.90 | | |

*Note. Regression coefficients and standard errors represent averaged values across all 100 bootstraps.*

to be a coping resource to avoid depression and anxiety [56]. Neither the CIs for the OSSS nor for having a steady partnership cover 0, so the effect can be considered significant.

Interestingly, higher age was associated with lower burden in the context of screening, which might reflect that a certain level of life experience might lower the negative consequences of screening. This predictor was selected for every model except for the physical dysfunction, however the CIs for the subscales are nearer to 0 than for the total PCQ scale.

For the health literacy scales, we found a positive relationship with the emotional and the social dysfunction subscales, though only the application of medical information subscale was selected. Health literacy is defined as the capacity to obtain, process, and understand basic health information. It is broadly investigated as protective factor for health outcomes and also the use of health care systems [57–59]. Thus, it is not surprising that it has a reducing effect on psychosocial outcomes of screening. The CIs lie away from 0, indicating that the found effect of health literacy is significant.

A look at the model fit suggests that our averaged models' RMSE values lie consequently beyond the scale deviation, which we interpret as well model fit for all four averaged predictions. The MAE values indicate that the average absolute error between the predicted and actual values is quite small in relation to the scale size across all models.

The results partly match to the existing body of literature. A study that applied the PCQ in the context of breast cancer screening found higher education, being unmarried, and having 2 or more children to be associated with lower levels of psychosocial distress [60]. Socioeconomic status is known to be correlated with subjective social status [61]. Though our models did not identify formal education (socioeconomic status) as relevant predictor, subjective social status was selected for all of our models. This suggests that the subjective perception of the status might be a better option to reflect the pattern behind the correlation than an objective measure.

A large Swedish study investigating the long-term development and predictors of psychosocial consequences of false-positive mammography screening found that early recall was the most influential predictor on outcomes measured by the COS-BC scale [62]. Furthermore, susceptibility, worry about breast cancer, dissatisfaction with information at recall and lack of social support were identified as relevant influential variables. Among the sociodemographic predictors investigated, the authors reported that country of origin was the most evident. Though not all results could be confirmed in our study, we could also identify social support, represented by two variables, as important predictor on psychosocial consequences of liver screening.

## Discussion

### Summary

The objective of this study was to identify relevant predictors for psychosocial consequences of a liver screening, which was implemented in the routine care of asymptomatic adults in Germany during the SEAL program phase. We identified three predictors that were selected both for the overall PCQ scale, as well as for its three subscales emotional, social and physical dysfunction. Among them, disease severity of comorbidities was the only one that had a positive, increasing effect on dysfunction. Higher subjective social status and social support had both a reducing effect on dysfunction on all three subscales and the overall scale. This finding emphasizes the social gradient, which was found for health outcomes in general and has to be considered relevant also in the context of liver screening. Our models also emphasize the role of social support, measured by the existence of a steady partnership or by the Oslo Social Support Scale, as an important factor to reduce psychosocial dysfunction. Age, which was found significant in three of the four models, might operate as a proxy for experience, which reduces dysfunction after screening. Health literacy in terms of the application of medical information was also found to be a protective factor against psychosocial dysfunction, though the effect was only identified for the emotional and the social subscale.

We suggest including all of them for further investigations on psychosocial consequences of liver screening to test their stability when using other data. It stands to question whether the results are transferable to other contexts of medical

screening, since we could only partially confirm the results of studies from other screening contexts. A systematic review on the psychosocial impacts of lung cancer screening identified age, socioeconomic status, informed decision-making and knowledge as well as social support as consistent predictors of psychosocial outcomes, which supports our findings [63]. However, gender, health-related beliefs, pre-existing burdens and patient experience were also identified as predictors, which our analysis did not confirm because the corresponding variables were not selected as relevant predictors. The effect of age was also identified in other screening contexts [14] as well as the effect of social support [62]. Further evidence is needed to test the general applicability and stability of the identified predictors.

## Limitations

This study shows some limitations that need to be discussed. Within the SEAL program, patients were screened over a longer period. Since psychosocial consequences were found to be varying depending on the time point measured, it is important to measure the outcomes after a fixed period for each patient, e.g., 2 weeks after screening. This procedure, however, requires a controlled clinical design, which was not realizable within the SEAL program. Due to administrative restrictions, we could not survey patients consecutively and instead had to survey them at a fixed time point. This approach had as result, that the instrument measured long-term effects for a majority of the population and short-term effects for an unknown part of the sample. In conclusion, the investigated outcomes here represent a mixture of both short- and long-term consequences of screening, which might limit the validity of the findings. In this study design, we also have to be aware that a certain degree of recall bias might occurs, the extent of which we cannot determine.

This study faces a further conceptual limitation, which is associated with its cross-sectional character. Due to the structure of the SEAL program, we were not able to implement a longitudinal design, which however, would have been more sophisticated to draw conclusions on causality of effects. Instead, we present correlations of variables that were measured at the same time. However, we think that this limitation is not severe, since many of the predictors can be considered relatively stable over time.

A limitation affecting both internal validity and generalizability of the results is the low response rate in our sample. While a sample size of 487 is sufficient for conducting statistical analyses, and bootstrapping was applied to enhance the robustness of predictors, it remains uncertain whether specific groups were underrepresented due to selection bias caused by non-response.

Another aspect of validity is related to the used instrument, the PCQ. Though it was implemented broadly in the past, criticism about the validity of the PCQ was raised. Due to the inconsistent findings in studies, which we also reported in the introduction, a group of researchers suggest that reasons might lie in the survey instrument itself [62,64]. Though the PCQ was considered a better alternative to measure psychosocial consequences as the generic measures Hospital Anxiety and Depression Scale, General Health Questionnaire and also the STAI, another instrument, the COS-BC, was developed and found to be more reliable and valid in measuring long-term consequences of screening in breast cancer [65,66]. It stands to question whether an adaption of the COS-BC would have produced other results. Our goal, however, was to apply an instrument that was easily adaptable to the context of screening for liver cirrhosis and fibrosis. The COS-BC is less generic and includes items that are specific to the context of breast cancer and thus, seemed not to be appropriate in our study. Furthermore, we wanted to discover whether findings from the context of liver screening match to the broad evidence from the field of cancer screening. For further detailed investigation of the specific psychosocial consequences of liver screening, we suggest to explore the potential of extension of the PCQ within qualitative studies like those that were done in course of the development of the COS-BC [66].

Within our study, we not only aimed to investigate psychosocial consequences of screening for liver cirrhosis in general, but also for the specific pattern of false-positive results. Since positive screening results generally appeared seldom within the SEAL program, and false-positive results occurred even more rarely, it was not possible to produce a database

that was large enough to systematically investigate the effect of false-positive results. In this context, we further have to discuss the relatively small return rate within this study. Due to limited funding, we could not implement incentives to increase the return rate. In our previous qualitative study, we gained the impression that not all participants were aware of their participation in the SEAL program. We therefore assume that the context of the survey was not clear to some people, which reduced return rates. However, the database was large enough to draw reasonable conclusions.

The identified predictors social support and having a steady partnership share a social component, which might be relevant within the patient-provider interaction. Poor patient outcomes are associated with bad patient-provider relationship [67]. Social support can act as a coping resource, when psychosocial strain is apparent [56]. The amount of social strain caused by the screening might be less dependent on the screening itself but more on the way the results are communicated within the patient-provider relationship. Our qualitative study, which we conducted in the context of the SEAL program, found that the time of uncertainty during waiting for the test results was a stressful component within the whole screening process [36]. In that study, all patients who reported negative emotions also reported limitations in the patient-provider relationship including a lack of trust. However, this result is not reflected in the predictor selection. Though we included variables on treatment evaluation, critical communication skills and information satisfaction, those did not seem to be important enough to be selected as predictors. It stands to question whether quantitative methods can measure such a narrative or whether the applied measurements are sensitive enough to represent the patient-provider communication.

Our models did not select the screening result (positive or negative) as relevant predictor for psychosocial outcomes. This result was surprising, since we expected a higher amount of social distress in the positive screening group. If a statistical effect is considered strong, it would also be reflected in a small sample size. Thus, we conclude that though there might be a systematic difference between the two groups, it was not strong enough to be detected by such a small sample size. On the other side, however, our sample composition reflects the real setup since the positive screening rate in SEAL was 4.12% [34].

## Strengths

To our knowledge, our study is the first study that investigated psychosocial consequences in the context of screening for liver cirrhosis and fibrosis. By this, we could demonstrate the applicability of the PCQ even in a non-cancer setting. Though we were not able to differentiate between short- and long-term consequences, we could identify a set of predictors that have an effect on the variance of psychosocial consequences after liver screening. For further investigations in this field, we recommend to measure the selected variables and further test their effect in a short- and long-term perspective. With our study, we add a systematic investigation of predictive structures of the PCQ that helps to understand the inconsistent findings reported in the literature.

## Conclusion

In general, the level of psychosocial consequences of screening were relatively low for our sample. The findings suggest, however, that how screening results and health information are communicated can significantly impact patient psychosocial health. Medical practitioners should consider personalized communication strategies that take into account the individual's health literacy, disease severity and social support systems. The protective role of social support and stable personal relationships suggests that integrating psychosocial support services within screening programs could mitigate negative outcomes. This might include referral to support groups or mental health services for those lacking robust personal support networks.

Furthermore, providing comprehensive educational materials to increase patient health literacy might help to demystify the screening process and its potential outcomes and can reduce psychosocial burden even if patients come from a segment of lower subjective social status.

## Supporting information

**S1 File. Authorship statement.**
(DOCX)

**S2 File. Original Study Questionnaire (positively screened patients).**
(PDF)

**S3 File. Original Study Questionnaire (negatively screened patients).**
(PDF)

## Acknowledgments

We thank all SEAL patients that filled out and returned the questionnaire. Furthermore, we want to acknowledge Matthias Sehlbrede for providing statistical guidance for this analysis. We also would like to express our thanks to the SEAL consortium for supporting our study within the SEAL program. Lastly, we want to thank Niklas Brunn for giving valuable feedback to the manuscript. The article processing charge was funded by the Baden-Wuerttemberg Ministry of Science, Research and Art and the University of Freiburg in the funding programme Open Access Publishing.

## Author contributions

**Conceptualization:** Urs Alexander Fichtner, Erik Farin-Glattacker.

**Data curation:** Urs Alexander Fichtner.

**Formal analysis:** Urs Alexander Fichtner.

**Funding acquisition:** Erik Farin-Glattacker.

**Investigation:** Urs Alexander Fichtner.

**Methodology:** Urs Alexander Fichtner.

**Project administration:** Urs Alexander Fichtner.

**Resources:** Urs Alexander Fichtner.

**Software:** Urs Alexander Fichtner.

**Supervision:** Andy Maun, Erik Farin-Glattacker.

**Validation:** Urs Alexander Fichtner.

**Visualization:** Urs Alexander Fichtner.

**Writing – original draft:** Urs Alexander Fichtner.

**Writing – review & editing:** Andy Maun, Erik Farin-Glattacker.

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
