## [Decision Letter · Decision Letter 0]

8 Aug 2024

PONE-D-24-19978Predictors for psychosocial consequences of screening for liver diseases: A data-driven approachPLOS ONE

Dear Dr. Fichtner,

Thank you for submitting your manuscript to PLOS ONE. After careful consideration, we feel that it has merit but does not fully meet PLOS ONE’s publication criteria as it currently stands. Therefore, we invite you to submit a revised version of the manuscript that addresses the points raised during the review process.

 The manuscript has been assessed by two reviewers and their comments are available below. The reviewers have raised some valuable discussion points and limitations of the study that need to be thoroughly discussed. Please review their comments and make the appropriate revisions. 

We look forward to receiving your revised manuscript.

Kind regards,

Emma Campbell, Ph.D

Staff Editor

PLOS ONE

Journal Requirements:

2. Thank you for stating the following financial disclosure: "funded by the Innovation Fund of Federal Joint Committee of Germany, provided by the Deutsches Zentrum für Luft- und Raumfahrt (Funding ID: 01NVF16026)".

Reviewers' comments:

Reviewer's Responses to Questions

**Comments to the Author**

1. Is the manuscript technically sound, and do the data support the conclusions?

Reviewer #1: Partly

Reviewer #2: Yes

2. Has the statistical analysis been performed appropriately and rigorously? 

Reviewer #1: I Don't Know

Reviewer #2: Yes

3. Have the authors made all data underlying the findings in their manuscript fully available?

Reviewer #1: Yes

Reviewer #2: Yes

4. Is the manuscript presented in an intelligible fashion and written in standard English?

Reviewer #1: Yes

Reviewer #2: Yes

5. Review Comments to the Author

Reviewer #1: ##How much time had passed since the subjects had been screened and then responded to the mailed surveys? Any concerns about recall difficulties or bias?

##I would be more convinced by a study which used a smaller sample but who more closely and prospectively tied their investigation of screening effects to the screening procedures themselves while utilizing mixed methods (qualitative interviews regarding patients' experiences with liver screening would be a valuable addition to the literature). The vast majority of these patients were screened negative and one would wonder what kind of negative psychology that would yield in them. If authors REALLY wanted to query psychological responses to liver disease screening, they should employ methods which investigate patient populations with more liver risks are present (i.e., more test subjects receiving positive screening results) and investigating patient populations where leading cirrhosis diagnoses (i.e., alcohol-related cirrhosis, Met-ALD, MASLD) are more likely to be present. As examples, conducting cirrhosis screening in patient populations seeking care in addiction or obesity clinics. They could also investigate the psychological impact of screening patients in the general population for excessive alcohol use--a key factor in liver damage and cirrhosis development in many countries. In my experience, discussion of excessive alcohol use and discussion of hazardous behaviors leading to liver disease (excessive drinking, overeating) ROUTINELY cause psychological distress in patients. I would also be interested in more granular detail about HOW patients received their screening data--who delivered the information and what kind of interaction took place as the results and implications were discussed.

##In sum, I wasn't surprised by the test results that in a sample comprised of largely persons receiving negative screening results about their livers and later asked to reflect on the psychosocial effects of this, that there was not much distress reported. I am certainly glad that screening for cirrhosis doesn't elicit distress--it is an important endeavor. Yet, my years of experience caring for at-risk liver patients, and reflections on what I've seen, suggests that the reality is likely to be quite different than this study's findings. For most people, receiving unflattering objective data about their liver health tied to health behaviors which people already tend to misrepresent to their physicians (diet, alcohol consumption) is often a distressing experience. In fact, it is this unpleasant psychological experience that liver clinicians and their psychosocial colleagues tend to harness therapeutically in order to nudge patients into making changes in their lifestyle to improve their liver health. Part of me reads this literature about the psychological effects of medical screening the way we might consider literature about the pain perceptions of patients undergoing IV placement--isn't it inevitable and perfectly normal that they will be uncomfortable to some degree?

##With so many psychometric instruments being used and without ready access to the items they contain and which domains were queried, it was difficult to parse what factors were influencing what. The domains were already rather nebulous as to how they were defined (e.g., what are the discrete postulated psychosocial consequences of medical screening?).

Reviewer #2: I read with interest the manuscript entitled "Predictors for psychosocial consequences of screening for liver diseases: A data-driven approach". I especially enjoyed the systematic approach in this study.

I have a few concerns and comments.

The study seems to lack data about the specific phase in which each participant was within the screening process. This apparently leads to the inability to study the consequences of false positive screening results.

Response rate was 9-12%, and apparently only 21 individuals with a positive screening result were included.

Another limitation is the assessment of a mixture of short- and long-term consequences of screening, since participants were at different stages of the screening process.

Large part of the text on pages 17-18 would be more appropriately located in the discussion

Was the screening result (pos vs neg) considered in the multivariable models? This should be clarified.

6. PLOS authors have the option to publish the peer review history of their article (what does this mean?). If published, this will include your full peer review and any attached files.

Reviewer #1: No

Reviewer #2: **Yes: **Fredrik Åberg

---

## [Author Response · Author response to Decision Letter 1]

9 Aug 2024

Dear Reviewers,

thank you for reviewing our article. We uploaded a point-by-point rebuttal letter within this submission. We hope we adressed all your points adequately.

---

## [Decision Letter · Decision Letter 1]

24 Sep 2024

PONE-D-24-19978R1Predictors for psychosocial consequences of screening for liver diseases: A data-driven approachPLOS ONE

Dear Dr. Fichtner,

Thank you for submitting your manuscript to PLOS ONE. After careful consideration, we feel that it has merit but does not fully meet PLOS ONE’s publication criteria as it currently stands. Therefore, we invite you to submit a revised version of the manuscript that addresses the points raised during the review process.

We understand that you have provided a waiver from your ethical approval committee for conducting the SEAL study. However, it appears as if the study described in your submission was not conceptualized as part of the original SEAL study. Therefore, we also require ethical approval or a waiver for the performed sub-study. Could you please upload the relevant document, and, if it is in German, an English translation.

We can see that you have previously published a study (https://doi.org/10.3389/fpsyg.2022.956674) using the same participant group, and that you have acknowledged this publication in your submitted study. Could you please comment on the study overlap in light of PLOS ONEs policy: We strongly discourage the unnecessary division of related work into separate manuscripts. We have also noted that in the above referenced publication the methods state that "we received 499 (negatively screened) respectively 21 (positively screened) completed questionnaires" while the current submission reads "our database for the analysis includes n=502 questionnaires from participants who were negatively screened and n=21 questionnaires from participants who were positively screened (net sample n=523)". Could you please comment on this difference?

We look forward to receiving your revised manuscript.

Kind regards,

Johanna Pruller, Ph.D.

Staff Editor

PLOS ONE

Journal Requirements:

Reviewers' comments:

Reviewer's Responses to Questions

**Comments to the Author**

1. If the authors have adequately addressed your comments raised in a previous round of review and you feel that this manuscript is now acceptable for publication, you may indicate that here to bypass the “Comments to the Author” section, enter your conflict of interest statement in the “Confidential to Editor” section, and submit your "Accept" recommendation.

Reviewer #1: All comments have been addressed

Reviewer #2: All comments have been addressed

2. Is the manuscript technically sound, and do the data support the conclusions?

Reviewer #1: Yes

Reviewer #2: Yes

3. Has the statistical analysis been performed appropriately and rigorously? 

Reviewer #1: I Don't Know

Reviewer #2: Yes

4. Have the authors made all data underlying the findings in their manuscript fully available?

Reviewer #1: Yes

Reviewer #2: Yes

5. Is the manuscript presented in an intelligible fashion and written in standard English?

Reviewer #1: Yes

Reviewer #2: Yes

6. Review Comments to the Author

Reviewer #1: (No Response)

Reviewer #2: All of my previous comments have been adequately addressed by the authors. Good job.

7. PLOS authors have the option to publish the peer review history of their article (what does this mean?). If published, this will include your full peer review and any attached files.

Reviewer #1: No

Reviewer #2: No

---

## [Author Response · Author response to Decision Letter 2]

17 Oct 2024

Please find our detailed statement in the rebuttal letter.

---

## [Decision Letter · Decision Letter 2]

16 Dec 2024

PONE-D-24-19978R2Predictors for psychosocial consequences of screening for liver diseases: A data-driven approachPLOS ONE

Dear Dr. Fichtner

Thank you for submitting your manuscript to PLOS ONE. After careful consideration, we feel that it has merit but does not fully meet PLOS ONE’s publication criteria as it currently stands. Therefore, we invite you to submit a revised version of the manuscript that addresses the points raised during the review process.

We look forward to receiving your revised manuscript.

Kind regards,

Khaled Hossain, Ph.D.

Academic Editor

PLOS ONE

Journal Requirements:

Reviewers' comments:

Reviewer's Responses to Questions

**Comments to the Author**

1. If the authors have adequately addressed your comments raised in a previous round of review and you feel that this manuscript is now acceptable for publication, you may indicate that here to bypass the “Comments to the Author” section, enter your conflict of interest statement in the “Confidential to Editor” section, and submit your "Accept" recommendation.

Reviewer #2: All comments have been addressed

Reviewer #3: (No Response)

2. Is the manuscript technically sound, and do the data support the conclusions?

Reviewer #2: Yes

Reviewer #3: Yes

3. Has the statistical analysis been performed appropriately and rigorously? 

Reviewer #2: Yes

Reviewer #3: Yes

4. Have the authors made all data underlying the findings in their manuscript fully available?

Reviewer #2: Yes

Reviewer #3: Yes

5. Is the manuscript presented in an intelligible fashion and written in standard English?

Reviewer #2: Yes

Reviewer #3: Yes

6. Review Comments to the Author

Reviewer #2: No further comments. Important work.

Reviewer #3: The manuscript entitled, “Predictors for psychosocial consequences of screening for liver diseases: A data-driven approach” investigated the predictors to explain variance in post-screening psychosocial consequences in the context of a liver screening for cirrhosis and fibrosis. They also try to establish the protective and negative impact variables. The manuscript is written in good English and I thought the flow of the manuscript needs to be improved. Most importantly, the study design and recruitment of study participants. This research should clarify the ethics and consent of study populations. Therefore, the current form of the manuscript needs to go through the revision. I appreciate the authors improving the manuscript, and to resubmit. I want to specify some major comments for improving the manuscript.

Abstract:

It would be appreciated if the authors showed the statistical values (for example, beta coefficient with 95% CIs) in the results sections. The conclusion part is too long in comparison to the results. It should be reduced by highlighting the specifications.

Introduction:

The study's main aim was to identify the psychosocial consequences of liver diseases, specifically liver cirrhosis and fibrosis. Although the study background states a lot of things, unfortunately, it misses out on the importance of psychosocial consequences of liver diseases that are well studied till now. It would be appreciated if the authors would consider the aforementioned point. Additionally, the objectives of the study are stated in a question sentence. It is better to avoid the question sentences even with numbering.

Methodology:

I want the authors to state the advantages of LASSO regression over other conventional regression analyses. Is it appropriate only for variable selection? Why did the authors conduct 100 bootstrapping? Did you conduct sensitivity analyses? If so, please specify.

Please state the participant's rights and confidentiality statement.

Results:

The overall response rate is too low. Does not it have the effect of analysis bias and overall findings?

Discussion and conclusion:

Strengths and limitations are merged in one section. The author should clarify the strengths and limitations separately.

The overall discussion is not up to the mark, it should be more specific in the context of the findings.

7. PLOS authors have the option to publish the peer review history of their article (what does this mean?). If published, this will include your full peer review and any attached files.

Reviewer #2: No

Reviewer #3: **Yes: **Dr. Nayan Chandra Mohanto

---

## [Author Response · Author response to Decision Letter 3]

28 Jan 2025

Dear Editor,

thank you for reviewing our manuscript again. Our detailed response to all your points is attached in the rebuttal letter.

With kind regards from Germany

---

## [Decision Letter · Decision Letter 3]

4 Feb 2025

Predictors for psychosocial consequences of screening for liver diseases: A data-driven approach

PONE-D-24-19978R3

Dear Dr. Fichtner,

We’re pleased to inform you that your manuscript has been judged scientifically suitable for publication and will be formally accepted for publication once it meets all outstanding technical requirements.

Kind regards,

Khaled Hossain, Ph.D.

Academic Editor

PLOS ONE

Additional Editor Comments (optional):

Reviewers' comments:

Reviewer's Responses to Questions

**Comments to the Author**

1. If the authors have adequately addressed your comments raised in a previous round of review and you feel that this manuscript is now acceptable for publication, you may indicate that here to bypass the “Comments to the Author” section, enter your conflict of interest statement in the “Confidential to Editor” section, and submit your "Accept" recommendation.

Reviewer #3: All comments have been addressed

2. Is the manuscript technically sound, and do the data support the conclusions?

Reviewer #3: Yes

3. Has the statistical analysis been performed appropriately and rigorously? 

Reviewer #3: Yes

4. Have the authors made all data underlying the findings in their manuscript fully available?

Reviewer #3: Yes

5. Is the manuscript presented in an intelligible fashion and written in standard English?

Reviewer #3: Yes

6. Review Comments to the Author

Reviewer #3: I am pleased on the revision made by the authors. They addressed all the comments appropriately. I have no further queries.

7. PLOS authors have the option to publish the peer review history of their article (what does this mean?). If published, this will include your full peer review and any attached files.

Reviewer #3: **Yes: **Nayan Chandra Mohanto

---

## [Editor Report · Acceptance letter]

PONE-D-24-19978R3

PLOS ONE

Dear Dr. Fichtner,

I'm pleased to inform you that your manuscript has been deemed suitable for publication in PLOS ONE. Congratulations! Your manuscript is now being handed over to our production team.

Kind regards,

on behalf of

Dr. Khaled Hossain

Academic Editor

PLOS ONE